# Spatial Problem-Solving in Working Dogs: The Combined Effect of Body-Size Awareness, Social Learning and Functional Breed Selection

**DOI:** 10.3390/ani16010060

**Published:** 2025-12-25

**Authors:** Petra Dobos, Péter Pongrácz

**Affiliations:** Department of Ethology, ELTE Eötvös Loránd University, Pázmány Péter sétány 1/c, 1117 Budapest, Hungary; dobpet2001@student.elte.hu

**Keywords:** detour task, functional selection, cooperative breeds, independent breeds, social learning, body awareness

## Abstract

Hundreds of recognized dog breeds show exceptional variety of morphology, and often markedly different behaviors, too. Based on the extent of their reliance on human-given signals, many dog breeds are characterized as either cooperative or independent workers. Here, we investigated the behavior of these two breed types in a complex problem-solving situation. We tested N = 149 dogs from more than fifty breeds, where the dogs had to obtain the reward by either making a detour around a transparent obstacle or using either an uncomfortably small or conveniently large opening as a shortcut across it. To add a social aspect to the test, in some groups, the experimenter showed the dogs how to detour the obstacle. We found that dogs made decisions that showed the effect of both body-awareness and social learning. Dogs chose the shortcut across the obstacle when it seemed comfortably large but opted for the longer detour especially when the opening was small and they observed the demonstration. Breed type had only a minimal effect: cooperative breeds spent more time watching the demonstrator. We conclude that flexible reliance of purebred working dogs on the available social and self-representation-based information could contribute to their success in the highly variable human environment.

## 1. Introduction

Dog breeds are human-made, cultural, and biological constructs. On the one hand, breeds are defined by their official description (the “standard”); on the other hand, a breed consists of specimens of a highly homogenous strain, characterized by breed-defining phenotypic traits and genomic makeup [1]. There are hundreds of dog breeds that were mainly developed from traditional landraces (e.g., [2,3]) or created through purpose-driven crossbreeding [1] over the last two hundred years [4]. Each modern dog breed has a unique “standard” that serves as a distinctive blueprint for not only the signature look, but also some of the behavioral traits of the dogs belonging to that breed [5]. There are grouping systems which help to sort the bewildering array of breeds into biologically relevant clusters, based on overarching ultimate or proximate processes of (mainly) artificial selection [6]. One of these systems capitalizes on the genetic relatedness among the breeds, thus creating so-called “clades” from the more closely related breeds (e.g., [7,8]). The other system is based on the more proximate process of functional breed selection. This resulted in dog breeds that can be sorted into function-specific clusters, depending on the tasks they were selected for by humans (e.g., [9,10]). The most often used functional categorization in ethology focuses on two main types: cooperative and independent working dog breeds (e.g., [11,12]). The main difference between them comes from the degree of the dogs’ dependency on human-given cues/instructions during work [9]. Herding dogs (e.g., Border Collie, Puli, Australian Shepherd) or various gundogs (such as the retrievers, spaniels, pointers) belong to the cooperatively working breeds. On the other hand, sled dogs (e.g., Siberian Husky), terriers and Dachshunds, sight- and scent-hounds (e.g., Whippet, Borzoi, Beagle, Bloodhound), and livestock-guarding breeds form the group of independently working dogs.

It is no surprise that independent and cooperative working breeds have been found to show markedly different responses in various dog–human interaction-based scenarios. For example, cooperative dogs more successfully follow human-provided visual pointing [9], establish eye contact with humans more easily [12], and perform more gaze-alternations during difficult tasks than independent dogs [13]. It was also found that independent dogs did not learn from observing the action of a human demonstrator in a detour task [14], and they were also generally less attentive to human verbal communication than cooperative breeds were [15]. However, the two breed types performed similarly in such tasks, where they did not have the opportunity to interact with, or rely on, human assistance [14,15,16,17]. This raises the question of whether the effect of functional breed selection truly manifests itself only in socially mediated tasks?

Problem-solving performance is not merely the function of human-dependent cognitive assets [18,19]. There are several intrinsic (age: [20]; sex: [21]; temperament: [22]; laterality: [23]; head shape: [24]; anxiety levels: [25]), extrinsic (training: [26]; life-history: [27,28]), and more complex factors (personality: [29]) which can affect dogs’ responses in various social and non-social scenarios. The recent division of various breeds into so-called “show” and “working” lines resulted in strong and predictable behavioural differences even within the same breeds [30,31]. On the other hand, one can hypothesize that any selective pressure that humans exerted on dog populations with a goal of achieving higher performance and reliability in a working task would result in basic behavioural similarities in these dogs (breeds) compared to other populations that did not undergo such selection. It was found, for example, that purposefully selected behavioural traits were more important in the success of explosive detection dogs than their morphology or sensory capacities [32,33]. A naturally occurring opportunity for showcasing such fundamental differences would be a comparison between purebred and mongrel dogs: most of the former are the result of strong artificial selection [34], while the latter are either the result of a random admixture of purebreds (e.g., [35]) or longer sessions of multi-generation reproduction of so-called “supermutts” (e.g., [36]). Here, it is important to note that a third cluster of mixed-breed dogs, the so-called designer crosses, should be cautiously treated in comparative studies because they are only a few generations removed from their parent breeds [37]. Therefore, the behavior of deliberately created designer crossbreds may show stronger resemblance to few selected purebreds [38] than to the multi-generation mixed breeds. In our paper, when we mention mixed-breed dogs, we always refer to the non-intentional, multi-generation mongrels.

There are sporadic indications that mixed-breed dogs are truly “different” in their socio-cognitive and task-related behaviors compared to purebreds. For example, mixed breeds were found to be less successful in following human pointing signals [9], and in a questionnaire study, according to their owners’ opinion, mixed breeds were harder to train than purebreds [39]. Another questionnaire study showed that according to the owners’ experience, mixed-breed dogs scored relatively high in aggression, and they showed the highest level of fear compared to purebreds [40]. Although different levels of prestige (appreciation by the owner and the resulting difference in care and effort to train) can be a source of bias in comparative studies on mixed-breed dogs and purebreds [39,41], we can hypothesize that there would be a smaller difference between different types of working dogs’ problem-solving behavior than between most of the purebreds and the mixed-breed dogs. To test this hypothesis, in this study, we opted for a complex testing protocol in which, recently, it was found that mixed-breed dogs show a highly flexible and optimized strategy of spatial problem-solving [42]. By using the same protocol as the researchers with the mixed-breed dogs did [42], we wanted to see whether purebred dogs from independent and cooperative types would behave similarly to mixed breeds, or if they would respond differently, perhaps because they have a different selection past regarding tasks that include interactions with humans.

The original spatial problem included a barrier-test, where dogs had to negotiate a straight, transparent obstacle to reach the reward on the other side [42]. Two solutions were offered, a longer detour and then a more optimal shortcut, which were eventually simultaneously available for the subjects. According to the protocol, at first, the dogs had to repeatedly detour the obstacle while the shortcut (an opening through the barrier) was shut. Additionally, dogs in one group could witness a (human) demonstrator who showed them how to detour effectively before the shortcut became available. To make the task even more complex, the opening (shortcut) was either comfortably large or adjusted to be just barely big enough for the subjects. In the case of the mixed-breed dogs, it was found that they preferred the shortcut over the detour but made a clear distinction between the small and comfortably large opening, thus proving that they relied on their body-size representation. According to the body-awareness-based shortcuts, mixed-breed dogs relied much less on the socially mediated information from the human demonstrator [42].

In our present experiment, we provided purebred dogs with a spatial problem, where we could test the potential effect of multiple intrinsic (breed type, body-awareness) and extrinsic factors (door size, demonstrator) on the dogs’ decision-making. We hypothesized that functional breed selection could influence how working dog breeds will behave in the complex spatial problem-solving task. Compared to the mixed breeds, we predicted that working dogs will show stronger reliance on the human demonstration, as they were selected for being more dependent on humans than the randomly bred mixed breeds [9]. Additionally, we predicted that the social learning effect will be the strongest in the cooperative breeds [14]. Regarding the body-awareness-based shortcuts (small vs. large door differentiation), we had two hypotheses. According to the first, we expected that as self-representation is a capacity that was most probably untouched by directional (functional) selection, purebred working dogs will show a similarly flexible reaction to the two types of openings as the mixed-breed dogs [43]. However, another hypothesis also seemed to be feasible: as purebred working dogs have been selected for higher task-related motivation [44], one could also predict that in their case, we will find less optimization for body size, and the working dog breeds will also negotiate the small opening without hesitation. Lastly, in this study, we avoided testing strongly brachycephalic breeds (dogs with high cephalic index) because earlier it was found that due to their forward-facing eyes and specific retinal structure [45], these dogs can show biased responses to stimuli positioned right in front of them (e.g., human pointing signals: [9]; choice of openings to enter: [46]). Nevertheless, we still took measurements of the subjects’ cephalic indices to be able to add this factor as a potential confounder to the analysis.

## 2. Methods and Materials

### 2.1. Subjects

Owners were recruited via advertisements on social media. We tested N = 149 purebred dogs that were at least 1 year old (see Appendix A). Both sexes, as well as intact or neutered/spayed dogs, were included (age: mean ± SD = 4.5 ± 2.8 year). We only included dogs from breeds that can be sorted either into the functional groups of cooperative working breeds or independent working breeds. We conducted the group-assignment of the breeds according to the official description of the breeds in the standards (as they appear at the FCI (Fédération Cynologique Internationale) https://www.fci.be/en/Nomenclature/, accessed on 22 December 2025). The same method of breed assignment was previously used in several other publications (e.g., [9,15]). We did not test companion breeds (“lapdogs”) with no clear work function. During the recruitment of the subjects, we were aiming to achieve the widest possible selection from both the cooperative and independent type, without over-representing any of the breeds. Thus, we tested a maximum of 3 dogs per breed. At the end, we tested N = 26 cooperative and N = 28 independent breeds. Because of the testing equipment, we had to apply some size restrictions during the recruitment of the subjects: we only tested dogs that fell between 30 and 90 cm height at the withers. The keeping conditions (indoor only; indoor–outdoor; outdoor only) and the training level of the dogs (none; training at home; dog school course; regular dog school; individual trainer; special sport/work training) were also recorded.

### 2.2. Exclusions

N = 21 dogs were excluded who could not detour the fence during the closed-door trials (N = 10 cooperative, N = 11 independent dogs). The data of the remaining N = 128 subjects were used for the statistical analysis.

### 2.3. Testing Equipment (A Device Identical to the One Used in the Study of Dobos and Pongrácz Was Used) [42]

We ran the tests outdoors at the University’s campus. The experimental device was a 1 m high and 3 m long transparent fence. We attached wire mesh, with a hole diameter of 20 mm, to a thin steel frame (Figure 1). When we set up this device, we attached one end to a tall chain link fence at the property border (the two fences met in a 90° angle). Dogs could not go around the property border fence because it was approximately 200 m long. It was also not possible for the subjects to dig under our device because of the steel frame. We used two digital cameras (Blow GoPro (San Mateo, CA, USA) and Panasonic (Kadoma, Japan)) on tripods to record each test for later behavioural analysis.

In the middle of the first 1 m wide section of the fence (i.e., the farthest from its free end), there was an adjustable opening (“door”) with five possible sizes. We could set the required door size by inserting a series of opening-reducers into the largest opening. The largest opening had an upwards swinging door on hinges, which was possible to lock in the lower position, thus creating the “closed door” condition.

Test groups were formed based on the door size, which the dogs were provided with. Thus, we had a group with a “large” door; this was comfortably big for the dog to pass through. The large door was at least as tall as the dogs’ height at their withers. In the other group, the dogs were faced with a “small but still suitable”-sized door, which was passable for the dogs, but they had to lower their posture for it. The smaller door’s height was always set as being shorter than the dog’s height at the withers by at least 11 cm, to a maximum of 20 cm. Table 1 shows how the dogs’ heights at the withers were paired with corresponding “large” or “small” door sizes.

### 2.4. Testing Procedure

(The general testing procedure, except for the added experimental groups’ “detour demonstration/small door”, was identical to what was used by Dobos and Pongrácz [42].)

At first, we informed the owners about the test; during this time, the dog was on a leash at the testing site. Then, the experimenter (always the same, 24-year-old female (P.D.)) measured the dog’s height at the withers with a foldable ruler. If it was necessary, we asked the owner to help hold the dog still for the height measurement. Additionally, the experimenter took a picture of each dog’s head from above, which was necessary for calculating the dog’s cephalic index.

For motivating the dogs to perform the task, we used either food or a toy, depending on the dog’s preference. We requested the owners to bring their dog’s favorite toy or treats with them for the test. To ensure that the dog was well-motivated in the chosen reward, before the actual testing would start, the experimenter gave a piece of treat to the dog from the plate we used for the experiment. In the case of toys, the experimenter briefly played with the dog by using the toy provided by the owner.

We formed four testing groups, where we aimed to have a balanced distribution of the subjects according to their sex, keeping, and training conditions. Each dog participated in one group only, and they were only tested once. At the beginning of each trial, the dog was positioned by the owner on the starting point (2 m from the fence). Based on this, the detour around the fence was approximately 7 m long, compared to the shorter route (going through the opening), which was only 2.1 m long.

### 2.5. Control Groups (Cooperative/Small Door N = 17; Cooperative/Large Door N = 15; Independent/Small Door N = 16; Independent/Large Door N = 15)

Each dog participated in 6 consecutive trials. At the beginning of the first trial, the experimenter was already on the opposite side of the fence (while she arrived there, we asked the owner to turn away with the dog; thus, the dog could not see as the experimenter detoured the fence). The experimenter called the dog’s attention by loudly saying the dog’s name and the word “Look!”, and at the same time she showed the plate with the food or the toy to the dog. Then, the experimenter put down the plate/toy on the ground, 15 cm from the fence. The experimenter stepped back 0.5 m, and then she signaled to the owner to release the dog. We allowed the owner to encourage the dog during the trial, but the owner had to remain at the starting point, and we told them not to use any directional commands or pointing cues.

The doors were always closed in the first 3 trials. Through the next three trials (Trials 4–6), we provided either a small or large opening to the dog, depending on the group designation of the subject. Each trial lasted for a maximum of 60 s, which was measured by the experimenter with a stopwatch. She started the stopwatch when the owner released the dog from the starting point. The trial ended when the dog reached the target, or after 1 min elapsed. Right before Trial 4, when the experimenter set the opening, we asked the owner to turn away with their dog while the E manipulated the opening.

### 2.6. Demonstration Groups (Cooperative/Small Door N = 17; Cooperative/Large Door N = 18; Independent/Small Door N = 15; Independent/Large Door N = 15)

Just like in the control groups, the dogs first participated in three closed-door trials, then three trials with either the large or the small door open. However, in each closed-door trial, at first the experimenter demonstrated the detour to the dog. The dog and owner were at the starting point, facing towards the fence. The experimenter started the detour from the starting point, heading towards the free end of the fence, carrying the reward (food plate or toy) in her hand. The whole time during the demonstration, the experimenter drew the dog’s attention to herself with ostensive verbal signals. After reaching the free end of the short fence, the experimenter turned in and walked along the fence until she arrived at its last section, where the door was. Here, the experimenter briefly held up the reward, then she put it to the ground, 15 cm from the door, and showed to the dog that her hands were empty. After this, the experimenter stepped back and then signaled to the owner to release the dog. At the end of each closed-door trial, the owner had to call the dog back and turn away with it from the fence; thus, the experimenter could return to the starting point without being seen by the dog. After the three trials with closed doors and demonstrations, we run the next three trials with either the large or the small door open, similarly to the control group.

### 2.7. Behavioural Coding and the Variables

For coding the videos, the BORIS version 9.6.4 software (© Olivier Friard and Marco Gamba) was used. Table 2 shows the description of the behavioural variables we used for the analysis. The reliability of the coding was checked with the help of an independent observer, who coded 20% of the videos (footage from 26 randomly chosen subjects). This analysis showed that our coding was reliable (reward latency: Cronbach’s alpha = 0.926; “Direction of approach”: Cronbach’s alpha = 0.819; “Looking at owner”: Cronbach’s alpha = 0.866; “Looking at experimenter”: Cronbach’s alpha = 0.712; “Looking at door”: Cronbach’s alpha = 0.882; “Demonstration watching in Trial 3” duration: Cronbach’s alpha = 0.999).

### 2.8. Calculation of the Cephalic Index (CI)

The head shape of the dogs can affect their visual performance with regard to the accuracy and ease of the central and peripheral vision [9,45,46]. Although we avoided testing strongly brachycephalic breeds, we wanted to add the head shape of the dogs as a potential confounder to the analysis. We used the photographs we took at the beginning of the experiment for calculating the dog’s CI. We used the ruler function of the GIMP image editing program 2.2.13. (http://www.gimp.org/), which gives the measurements in pixels. The width of the dog’s skull was measured between the two zygomatic arches, while the skull length was measured between the tip of the nose and the occipital protuberance. Then, we divided the head’s width by the head’s length and multiplied the ratio by 100, which yielded the CI. The larger the CI was, the shorter the skull of the subject was.

### 2.9. Statistical Analysis

Raw data for this research are available at Mendeley Data (Pongrácz, Péter (2025), “Complex door or detour/purebred dogs”, Mendeley Data, V1, doi: 10.17632/mpf32rw24p.1 [47]). All the statistical analyses were performed with the IBM SPSS.29 software. We added biologically meaningful two-way interactions to the initial models; then, one-by-one, we removed each non-significant interaction (backward model selection). We always report the results of the final (simplest) model. The significance level was α = 0.05.

The normal distribution of our data was checked by visual inspection of the Q-Q plots of residuals (relative frequencies, durations, and latency-type data). We analyzed latencies separately in the case of the closed- and open-door trials. Generalized Linear Mixed Models (GLMMs) were used, with trials as the repeated factor, and breed type, dogs’ sex, keeping, training, CI, and type of demonstration (control vs. demo) were the fixed factors. When analyzing the latencies of the open-door trials, door size was also added as a fixed factor.

Relative frequencies and durations (looking at the owner, at the experimenter, and at the door; frequency of the attempts made to get through the door) were analyzed with GLMMs, where we used the breed type, door size, demo type, and CI as fixed and trials as repeated factors. Additionally, training and keeping were included in the models in the case of looking at the owner/experimenter.

Generalized Estimating Equations (GEEs) with ordinal logistics were used for analyzing the solution choice (Trials 4–6, and separately in Trial 4 only). The dog’s ID was the random factor, trials were the repeated factor, and door size, breed type, demo type, and CI were added as fixed factors. Approach direction (diagonal vs. straight) was analyzed with GEE with ordinal logistics. We compared the first three and the last three trials separately. Trial was the repeated factor; we also added CI, breed type, demo, and door size (in the case of Trials 4–6) as fixed factors to the model.

The relative duration of watching the demonstration in Trials 1–3 was analyzed in connection with solution choice in Trial 4 and breed type as fixed factors with GLMM. We compared the CI of dogs (as a continuous variable) across the testing groups with the Generalized Linear Model (GLM), where experimental groups, breed type, demonstration, and door size were the fixed factors.

## 3. Results

First, we analyzed the distribution of CIs across the experimental groups (Figure 2). Our sample had a balanced distribution of the head shape of the dogs, as CI did not associate significantly with the experimental groups (*F*_1,117_ = 0.711; *p* = 0.663).

We examined whether the dogs chose to detour the fence or use the door in the open-door trials. The solution choice had a significant association with the size of the door (χ^2^_1_ = 6.795; *p* = 0.009) and with trials (χ^2^_2_ = 7.637; *p* = 0.022). The dogs chose to detour the fence more often in the case of the small door (Figure 3). The dogs chose the door more often in the last trial than in Trial 4, when the doors were open for the first time. We did not find a significant association between the choice of solution and the other fixed factors (CI: χ^2^_1_ = 0.032; *p* = 0.858; breed type: χ^2^_1_ = 0.056; *p* = 0.813; demo: χ^2^_1_ = 0.689; *p* = 0.407).

We analyzed separately the solution choice in Trial 4, when the dogs were met with the open doors for the first time. The effect of cephalic index and door size showed a significant interaction (χ^2^_2_ = 6.101; *p* = 0.014). Shorter-headed dogs preferred to go through the large door, and they would rather detour if the door was small (Figure 4). Longer-headed dogs chose between the solutions similarly regardless of the door size. We found a significant interaction between the cephalic index and the demonstration, too (χ^2^_2_ = 6.517; *p* = 0.011). Shorter-headed dogs with demonstration in the previous trials were more likely to choose to detour in Trial 4. The breed type did not affect the choice (χ^2^_2_ = 0.049; *p* = 0.825).

We analyzed whether watching the demonstrator in Trial 3 was associated with solution choice in Trial 4, or with breed type. We found a significant breed type effect (*F*_1,61_ = 6.536; *p* = 0.013): cooperative dogs watched the demonstrator for significantly longer (Figure 5). The relative duration of watching the demonstration was not associated significantly with solution choice in Trial 4 (*F*_1,61_ = 0.130; *p* = 0.720).

To analyze the dogs’ direction of approach (whether they ran straight towards the door or chose to run towards the free end of the fence), we compared separately the first three and the last three trials. Trial had a significant association with approach direction in the closed-door trials (χ^2^_2_ = 32.011; *p* < 0.001). The dogs were more likely to run straight ahead in Trial 1, and then they were more likely to run towards the end of the fence (Figure 6a). Moreover, we found a strong effect of demonstration (χ^2^_2_ = 14.896; *p* < 0.001). The dogs in the demonstration groups learned to follow the demonstrator, as they significantly more likely ran in the diagonal direction than the dogs in the control groups (Figure 6b). The CI of the dogs (χ^2^_1_ = 0.002; *p* = 0.968) and the breed type (χ^2^_1_ = 0.778; *p* = 0.378) did not affect their approach directions.

On the other hand, a significant association was found in the open-door trials between the approach direction and the size of the door (χ^2^_2_ = 4.485; *p* = 0.034). The dogs preferred to run diagonally towards the free end of the fence when the small door was open (Figure 6c). Demonstration also had a main effect; when the dogs observed the demonstration in the previous three trials, they made more diagonal approaches (χ^2^_2_ = 5.792; *p* = 0.016) (Figure 6d). Other factors such as the breed type (χ^2^_2_ = 0.000; *p* = 0.984), the CI (χ^2^_2_ = 0.006; *p* = 0.936), or repeated trials (χ^2^_2_ = 0.327; *p* = 0.849) did not affect the results.

In the case of reward latencies, we found a significant interaction of demonstration and trials in the first three (closed-door) trials (*F*_2,22_ = 3.597; *p* = 0.045). The dogs in the demonstration groups reached the reward faster from trial to trial (Figure 7). We also found that cephalic index (*F*_73,11_ = 2.669; *p* = 0.038) and training (*F*_5,11_ = 3.794; *p* = 0.030) had a significant effect. Shorter-headed dogs reached the reward faster than longer-headed dogs did (Figure 8). According to Tukey’s post hoc test, there was no significant difference between the various training levels; we found only a trend-like effect, where dogs with special training were somewhat faster than dogs that only participated in dog school classes. Other fixed factors did not have a significant effect (door (*F*_1,11_ = 1.583; *p* = 0.234); breed type (*F*_1,11_ = 0.090; *p* = 0.769); neutered status (*F*_1,11_ = 0.490; *p* = 0.499); sex (*F*_1,11_ = 0.103; *p* = 0.754); keeping (*F*_1,11_ = 1.633; *p* = 0.228)).

In the open-door trials (Trials 4–6), the detour demonstration (*F*_1,9_ = 1.970; *p* = 0.194), breed type (*F*_1,9_ = 1.514; *p* = 0.250), and door size (*F*_1,9_ = 0.021; *p* = 0.887) did not have a significant association with reward latency. This means that once the door was open, the dogs ran at the same speed to obtain the reward. Other fixed factors (CI (*F*_72,9_ = 1.133; *p* = 0.455), sex (*F*_1,9_ = 0.418; *p* = 0.534), neutered status (*F*_1,9_ = 0.021; *p* = 0.734), keeping (*F*_1,9_ = 1.467; *p* = 0.257), or training (*F*_1,9_ = 1.680; *p* = 0.235)) did not have a significant effect on the reward latencies either.

In the case of looking at the owner, we found a significant association with dogs’ sex (*F*_1,9_ = 5.753; *p* = 0.040): male dogs looked slightly more often at their owner than females did (see all the results in Appendix A). However, in the case of looking at the experimenter, we only found a significant trial effect (*F*_5,45_ = 3.373; *p* = 0.011): the dogs looked less at the experimenter as the trials went on (see all the results in Appendix A). Similar results were found regarding looking at the door (*F*_5,45_ = 4.139; *p* = 0.004): the dogs looked less at the door as the trials went on (see all the results in Appendix A). Finally, in the case of the dogs’ frequency of trying to get through the door (successful and unsuccessful attempts), no significant associations were found (see all the results in Appendix A).

## 4. Discussion

We examined purebred dogs that were selected for different types of work in a body-awareness-based spatial task with an additional chance for learning from a human demonstrator how to perform an effective detour. Our results confirmed that like mixed-breed dogs [42], purebreds also rely on body-size awareness when they have the option to choose between qualitatively differing solutions in a spatial problem-solving task. In the open-door trials where the door was slightly uncomfortable compared to the dogs’ height, the dogs opted more frequently for the detour than in the trials with the large door. Meanwhile, if the door was conveniently large, they would choose this optimal, shorter solution. Remarkably, in this current task, the “small” door was still big enough for the dogs to pass through without any extraordinary difficulty (i.e., they did not need to crawl to get through the small door). This feature was different in comparison to earlier studies [43,46], where the hesitation of the dogs before they approached the smallest doors was only observed when these openings were physically too small for the dogs. From this aspect, our current results provide an interesting new insight into dogs’ body-size awareness, because we found that dogs may decide to avoid smaller apertures, even if these offer only a slight inconvenience. Notably, our current experimental design offered an alternative solution to the dogs (the detour), which is a difference compared to the study of Lenkei et al. [43], where the dogs could only overcome the obstacle via a single, gradually decreasing-in-size door. Our results can be parallelled with the behavior of cats, who hesitated when they were offered gradually shorter openings on a barrier and started to look for other solutions (e.g., jumping over the barrier) [48]. The length of the detour and the suitability of a potentially available alternative solution were found to be important factors of decision-making in other species as well (ants: [49]; toads: [50]; cats: [51]). In our current experiment, the detour was more than three times longer than the shortcut via the opening to the reward, and according to the results, dogs optimized their choices based on their body-awareness (preference for shortcuts in the case of large door vs. relative preference for detour in the case of small door and socially acquired reinforcement of making detours).

Approach direction was also a telltale sign of the initial choice dogs made while negotiating the obstacle. When dogs encountered a small door, they more often ran diagonally towards the free end of the fence (i.e., more likely decided to make a detour in this scenario). These results show that dogs a priori assessed the size of the available opening compared to their own size. Therefore, our hypothesis that there would be a weaker difference between the purebred working dogs’ problem-solving choices (i.e., opting for the door or a detour) than in the case of the mixed-breed dogs can be rejected. Our other expectation, that self-representation is a capacity that most probably remained untouched by artificial (functional) selection, seems to be a more adequate explanation. The very mild difference we found between these dog populations (purebred vs. mixed-breed; cooperative vs. independent working dogs) could also imply that in complex, ecologically valid scenarios, the cumulative effect of different variables (breed, social learning, body-awareness) could have a balancing effect. As a result, in a realistic setup, we ended up with the “average dog behavior” in our sample. The effect of one or the other factor can be entangled and detected in experiments, where the testing design directly focuses on specific variables. For example, strong effects of functional breed selection were detected, among others, in scenarios of visual communication [9,13] and social learning [14,16].

When comparing the results of purebred working dogs to the earlier study on mixed-breed subjects [42], it is important to note that for mixed-breed dogs, three experimental groups were enough to reliably detect the effect of body-awareness. In the case of purebred dogs, we opted for the full factorial design, which also included the small door-demonstration condition (which was missing from the study with mixed breeds). In this condition, the effect of door size (“uncomfortably small”) and social information (detour demonstration) both prompt the dogs to choose the same solution: opt for detouring the fence. Our current results with purebred dogs showed that it was exactly the combination of small door size and detour demonstration that most intensely elicited body size-related decisions (regarding the solution choice and approach direction). This allows us to formulate an interesting new hypothesis, which would imply that purebred working dogs might have been selected to be more task-oriented, while (non-intentionally bred, multi-generation) mongrel dogs may have adapted to a great extent to thrive and reproduce without human assistance [1]. Consequently, in the case of mixed breeds, it is easier to detect body-awareness (the dogs are being more “cautious”), while the various working dog breeds need an additional factor, social learning through detour demonstrations, to abandon the option of choosing the smaller doors. As evolutionary support to this theory, we should note that most exemplars for the reliance on body-size awareness come from such species where collisions with or getting stuck in obstacles represents a danger (e.g., bumblebees; snakes; ferrets and budgerigars: see for a review [52]). However, without further convincing experiments, the body-awareness capacity of purebred working dogs and mixed-breed dogs can be considered apparently similar, and the effect of functional breed selection might manifest itself in socially mediated tasks only. It is also possible that even stronger differences could be detected between the body-awareness of purebreds and free-ranging dogs, who, unlike the mixed-breed dogs in our previous study [42], are reproducing and surviving without almost any human assistance [53].

In this study, with the purebred working dogs, we found stronger evidence of relying on the socially gained information than we did in the case of mixed-breed dogs [42]. Dogs that observed the demonstration were faster in learning the task than the subjects in the no-demo control groups, as their performance significantly improved from trial to trial. The dogs in the demonstration groups also decided to make a detour at the beginning of the trials, as they approached the fence diagonally when the doors were closed, which also could help them make their detours faster and more effectively. Moreover, social learning affected their direction of approach in the open-door trials as well: dogs still more often started to run diagonally towards the end of the fence in the demonstration groups even if the doors were available. Interestingly, these dogs most probably changed their mind later and eventually used the door as there was no difference between the demonstration groups and the control groups regarding their solution choices in the open-door trials. Therefore, our prediction that working dogs will show stronger reliance on human demonstration and they will keep detouring even if the door is open, because they were selected to be more dependent/obedient towards humans [9,13,54], was only partly supported.

Additionally, we also predicted that the effect of social learning would be strongest in the cooperative breeds [14,15]. However, in this study, the cooperative and independent working breeds behaved similarly in most aspects. One possible explanation could be that in the earlier detour experiments, where strong differences were found between the social learning performance of independent and cooperative dogs [15,16], the subjects had to negotiate a difficult V-shaped obstacle. In our current experiment, the dogs had to detour a straight obstacle, and it was shown by Pongrácz and Veres [55] that dogs had much less difficulty with detours around a straight fence compared to the V-shaped one. Nevertheless, we still detected in this experiment that cooperative breeds spent more time watching the demonstration, which falls in line exactly with our previous findings regarding the breed type-dependent difference in dogs’ attentiveness towards human actions [9,15,56].

Moreover, we found that dogs with different head shapes showed somewhat different responses during spatial problem-solving, like what was found in a previous study [46]. Shorter-headed dogs detoured faster when the doors were closed, which implies that due to their different visual capacity [45], they might see better that the doors are closed and there is a barrier in front of them. It was found that ganglion cells concentrate more in a central area of the retina in shorter-headed dogs than in longer-headed ones [45], which resulted in a more efficient performance of brachycephalic breeds in visual communication with humans [9,12]. In our current study, shorter-headed dogs optimized their behavior immediately in the first open-door trial and detoured if the door was slightly small; however, they went through it if it was comfortably large, which also supports the theory that shorter-headed dogs can more effectively assess the details of their environment when it is right in front of them. Longer-headed dogs might have better peripheral vision; therefore, potentially, they could not estimate the size of the door as precisely/easily in a split second as shorter-headed dogs did. These initial differences in spatial problem-solving between dogs with slightly different head shapes quickly diminished as the trials went on. Another related finding was that in our study, the dogs’ cephalic index did not show an association with the frequencies of looking at the owner or experimenter; meanwhile, in earlier studies, it was found that shorter-headed dogs more easily established eye contact [12] and gazed at human participants [24] more than the longer-headed dogs did. This difference compared to earlier publications can be explained with the easier task we provided to the subjects compared to the “impossible task” in the study of Ujfalussy and colleagues [24], and with the fact that we did not have extreme brachycephalic breeds in our sample (such as Pugs, Bulldogs, or Cavalier King Charles spaniels).

### Limitations to the Study

Although we recruited our subjects from dog breeds that were once selected for performing various working tasks, nowadays, most purebred dogs are primarily bred and kept as companions. One could suggest that this may result in diminishing behavioural differences between the once distinct categories of cooperative and independently working dog breeds. However, we should note that recent studies that found strong differences between the human-directed behavior of cooperative and independent breeds (e.g., [9,13,14,15,16]) used subjects from the same dog population as the present investigation. This makes it unlikely that the lack of pronounced breed type effect in our current study would be the result of today’s tendency of breeding dogs mostly for family dog purposes, which might weaken their original behavioural traits when they become “averaged out”-temperament companions.

Another factor that we did not test here was the familiarity of the demonstrator. In earlier experiments that measured dogs’ attentiveness and referential looking towards the nearby human, researchers found that dogs differentiated between their owner and a stranger [57,58]. However, in experiments where dogs had the opportunity to learn how to make a detour from either the owner or a stranger, no difference has been found between the effect of the demonstrators on dogs’ social learning [59,60].

## 5. Conclusions

In our body-awareness-based problem-solving task, which also included the possibility for social learning, we found that purebred working dogs show flexible and optimized behavior. Compared to earlier studies with functionally selected (independent and cooperative) working breed types [13,14,15,16], here we found only a negligible effect of breed type on the problem-solving behavior of the dogs. However, we found that extrinsic factors (door size; the availability of a demonstrator) had a much stronger influence on the decision-making of the subjects. There can be multiple reasons why functional breed selection did not affect relevantly the behavior of dogs in this scenario. One explanation might be that in such complex tasks as the one we used here (detour with or without demonstration, plus a shortcut with more or less convenient door size), flexible behavior provides the strongest advantage, and such flexibility can be achieved by learning (i.e., body-awareness and social learning) rather than by directional selection [61]. According to a different hypothesis, functional selection could result in such task-oriented breeds that respond to problem-solving situations with a uniformly high level of drive and persistence, which can mask their different responsiveness to human-given cues [15]. The relatively easy task (making a detour around a straight obstacle, using a shortcut) could also contribute to this outcome, as it was found earlier that cooperative and independent breeds did not differ in their persistence and capabilities in individual problem-solving tasks [13,54]. Our experiment convincingly showed that working dog breeds, as well as mixed-breed dogs in the earlier study [42], respond with high flexibility and resourcefulness to complex spatio-social tasks. This adaptive, learning-based performance could be an important asset to the general success of dogs in their adaptation to an extremely variable and complex anthropogenic environment.

## Figures and Tables

**Figure 1 animals-16-00060-f001:**
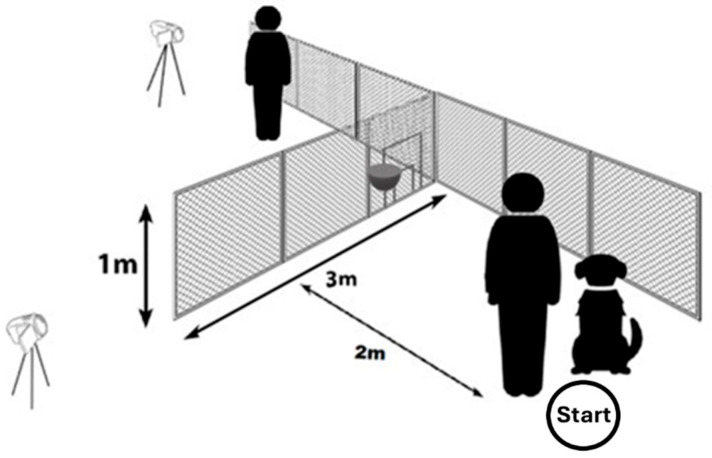
The experimental setup with the drawing of the transparent obstacle, which was fastened to the property border fence. The two cameras on the tripods are also visible. The outlay of the setup is identical to the one used by Dobos and Pongrácz [42]. The owner and the dog are standing on the starting point, while the experimenter stands on the opposite side of the fence. We depicted the small and large doors in a closed position. The small plate behind the doors was used for the reward.

**Figure 2 animals-16-00060-f002:**
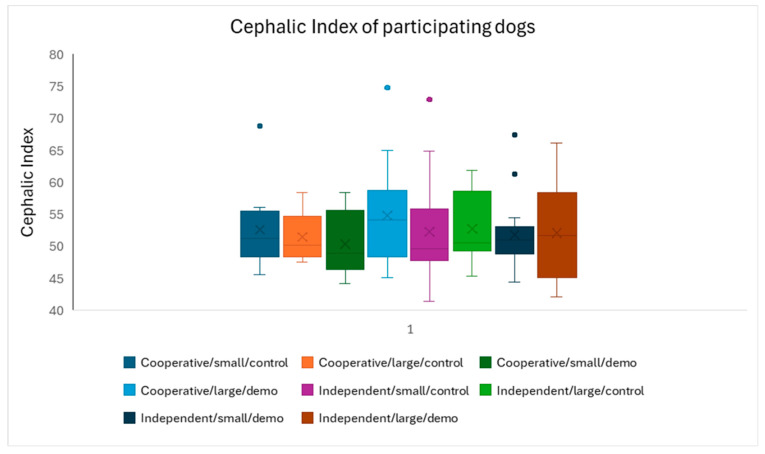
The cephalic index of dogs did not differ across the experimental groups. Cooperative and independent = breed type; small and large = door size. Box = inter-quartile (50%) range; whiskers = upper and lower quartile; horizontal line = median; X = average.

**Figure 3 animals-16-00060-f003:**
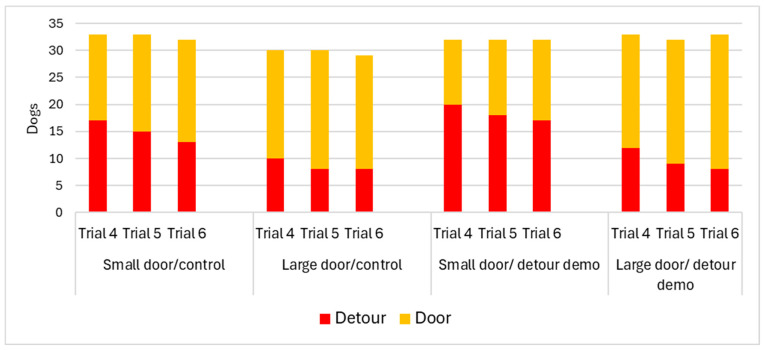
Solution choice during the open-door trials (Trials 4–6) in the experimental groups. Dogs opted more often to make detours when they faced the small door, and in Trial 4, before repeated exposure to the open doors.

**Figure 4 animals-16-00060-f004:**
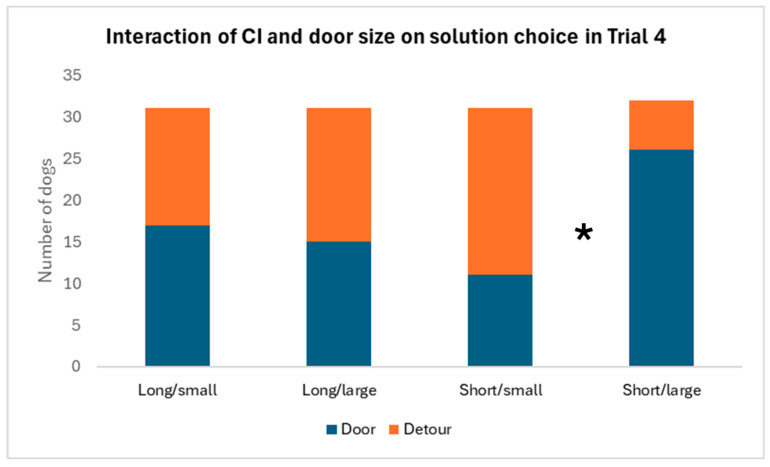
Solution choice in Trial 4, regarding the interaction between cephalic index (long or short) of the dogs and the size of the door (small or large). Shorter-headed dogs made a clearer distinction between the “small” and “large” door when they first met with it and more likely opted for a detour when the door was small. For a better visualization of the results, we divided the otherwise continuous cephalic index score at CI = 50.9, resulting in two halves of the sample. * = *p* < 0.05.

**Figure 5 animals-16-00060-f005:**
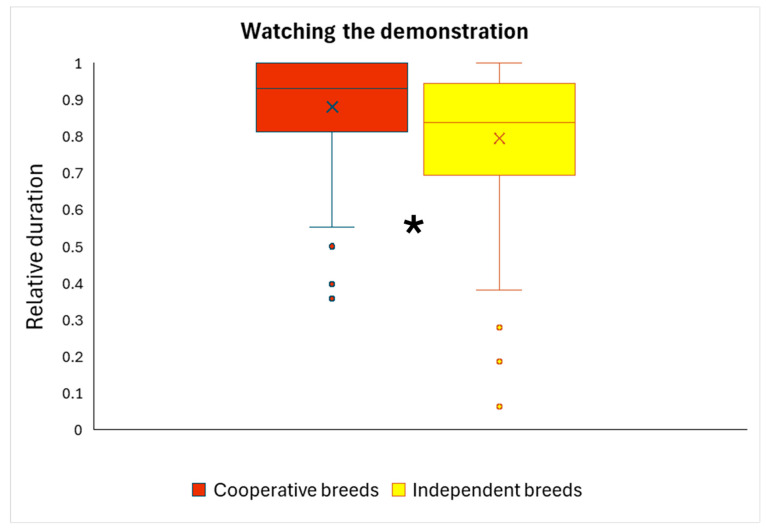
Watching the demonstration according to breed types. Cooperative working dogs watched the demonstrator for significantly longer in Trial 3. Box = inter-quartile (50%) range; whiskers = upper and lower quartile; horizontal line = median; X = average. * = *p* < 0.05.

**Figure 6 animals-16-00060-f006:**
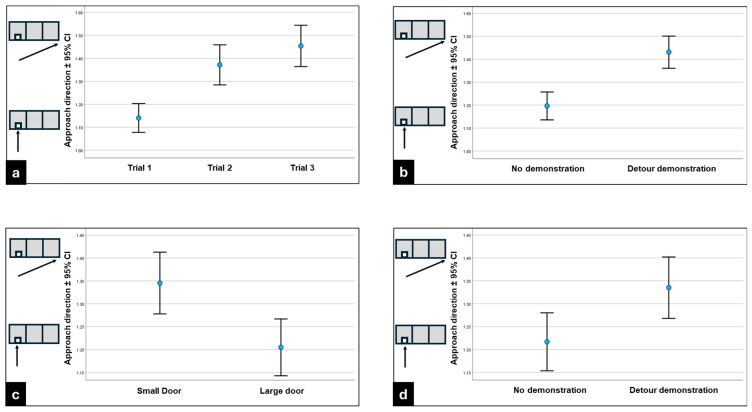
(**a**–**d**) Direction of approaching the fence. Small directional drawings besides the vertical axis help to understand the meaning of the values. Vertical arrow: the dog approached the opening directly; Oblique arrow: the dog approached the end of the fence first. (**a**) In the closed-door trials, dogs increasingly readily started to detour the fence along the trials. (**b**) In the closed-door trials, when they observed the demonstrator, the dogs more effectively made a detour. (**c**) In the open-door trials, the dogs started to detour the fence more likely when they were facing the small door. (**d**) In the open-door trials, the dogs more likely started to detour the fence when they observed the demonstration in the previous, closed-door trials. CI = confidence interval.

**Figure 7 animals-16-00060-f007:**
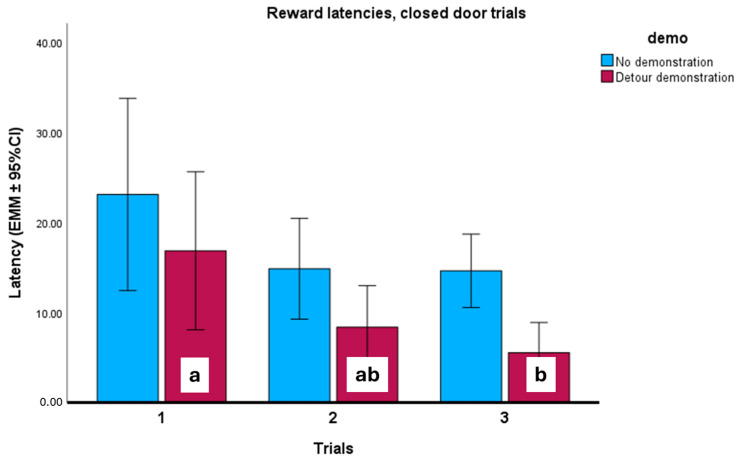
Reward latencies during the closed-door trials (Trials 1–3) in the groups with and without demonstration. In the demonstration groups, the dogs made detours significantly faster. Different letters mark a significant difference between the trials with detour demonstration. EMMs = estimated marginal means; CI = confidence interval.

**Figure 8 animals-16-00060-f008:**
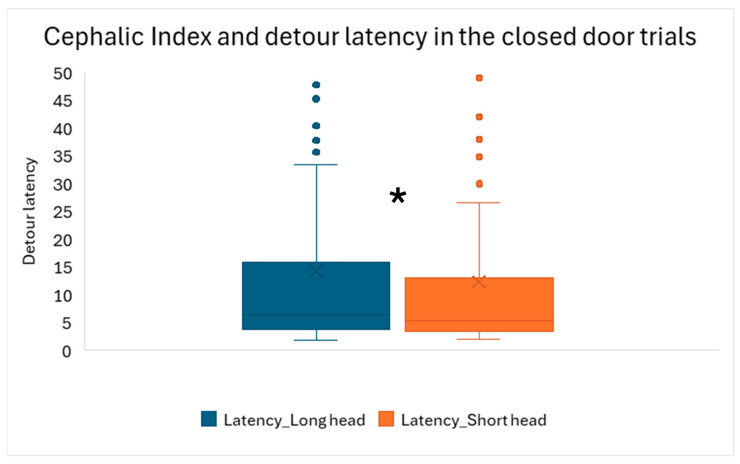
Reward latencies during the closed-door trials (Trials 1–3) in connection with the cephalic index (CI) of the dogs. Shorter-headed dogs made significantly faster detours than the longer-headed dogs did. For a better visualization of the results, we divided the otherwise continuous cephalic index score at CI = 50.9, resulting in two halves of the sample. Box = inter-quartile (50%) range; whiskers = upper and lower quartile; horizontal line = median; X = average. * = *p*<0.05.

**Table 1 animals-16-00060-t001:** Dogs’ height at the withers with the corresponding heights of small and large doors. Measurements are in cm.

Dog	Small Door	Large Door
71–90 (N = 4)	70	-
61–70 (N = 29)	50	70
51–60 (N = 40)	40	70
41–50 (N = 25)	30	50
30–40 (N = 30)	20	40

**Table 2 animals-16-00060-t002:** The behavioural variables and their descriptions (based on [42]).

Behavioural Variable	Unit	Description	
Success	Occurrence (0/1)	The dog reaches the target (eats the food, touches the toy).	
Choice (Trials 4–6 only)	Door/detour	The solution used by the dog for reaching the target.	
Reward latency	(s)	Time elapsed between the moment of releasing the dog at the starting point and the dog’s arrival at the reward (either by performing a detour or using the door).	
Looking at the door	Relative duration	The dog turns its head towards the opening (total duration/reward latency)	
Looking at the owner	1/s	The dog turns towards the owner (either by its head, or with its whole body). The *number of looking events* is divided by the *reward latency*.	
Looking at the experimenter	1/s	The dog looks at the experimenter (on the other side of the fence). The *number of looking events* is divided by the *reward latency*.	
Direction of approach	Straight or diagonal	The dog can either run towards the door (“straight”) or it can start running towards the free end of the fence (“diagonal”).	
Watching the demonstration (in Trials 1–3)	Relative duration	The dog watches the experimenter while she demonstrates the detour (*total duration of watching/duration of the demonstration*).	

## Data Availability

A file with the behavioural data and the fixed factors used for the statistical analysis is available at Pongrácz, Péter (2025), “Complex door or detour/purebred dogs”, Mendeley Data, V1, doi: 10.17632/mpf32rw24p.1.

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
