# Peer review of "Spatial Problem-Solving in Working Dogs: The Combined Effect of Body-Size Awareness, Social Learning and Functional Breed Selection"

_animals, 2025, doi:10.3390/ani16010060_

Round 1

Reviewer 1 Report

Comments and Suggestions for Authors

Overall:  Overall this is an interesting study looking at different working breeds and how they may differ in their decision making when humans are involved. 

Simple Summary and Abstract: Overall, very good.  The simple summary should have the number of dogs (n=) in addition to the number of breeds.  The abstract needs some rewording for clarification. See below.

Line 21: delete the word “now”

Lines 34-35: Due to how the sentence is structured, no question mark is needed. 

Lines 35-38: This sentence and description are confusing.  Please reword to clarify. 

Introduction:  Overall the introduction is good and provides sufficient background references for the subject.  Additional clarification is needed in places, however.  See below for suggestions. 

Lines 59 and 61: These sentence should begin with the category of breed rather than breed examples.  This change would help with the flow of the paragraph.

Lines 116-119: Break this sentence into two sentence to improve clarity of description.

Materials and Methods:  Overall, good.  Although it is mentioned early in the testing procedure, it would help to continue to clarify that the the methods described are identical to the study by Dobos and Pongracz (use citations).

Line 155: Please clarify from where the standards are sourced and how the classification is categorized (meaning what organization or authority provides the classification and distinction of breeds that was used to sort the dogs for this study).

Lines 209-213: Please provide reasons for the experimenter to play with the dog or give the dog treats prior to the experiment. 

Lines 232-238: In the control group description, the doors were closed for the first three trials.  Please clarify why the closure or opening of the doors was not varied in the control group.

Section 2.8: Please clarify why the CI is important for this study.

Results:  Overall the results are clear.  The statistical analyses were appropriate and the findings were presented in standard form. Visuals helped to clarify the findings and improve visualization of the results.

Discussion:  The discussion, limitations, and conclusions are appropriate and well written.  The discussion includes a lot more detail about brachycephalic breeds and justification for including CI.  I recommend including some of this in the introduction to help justify the measurements associated with CI and why it’s important. 

The Limitations section should also discuss the difference between the experimenter being the social learning variable versus what might happen if the owner played this role. 

Author Response

Responses to REVIEWER 1

Overall:  Overall this is an interesting study looking at different working breeds and how they may differ in their decision making when humans are involved. 

RESPONSE: We are grateful to the Reviewer for the supportive attitude and helpful comments. We did our best to answer your questions and incorporate the suggested changes to the manuscript. We hope that as a result, the new version of our paper becomes clearer and more impactful this way. 

Simple Summary and Abstract: Overall, very good.  The simple summary should have the number of dogs (n=) in addition to the number of breeds.  The abstract needs some rewording for clarification. See below.

Line 21: delete the word “now”

Lines 34-35: Due to how the sentence is structured, no question mark is needed. 

RESPONSE: Thank you for these suggestions, we corrected the text according to the Reviewer’s advice.

Lines 35-38: This sentence and description are confusing.  Please reword to clarify. 

RESPONSE: Thank you for the request. We changed the wording of this section, this is how it reads now (lines 35-37):
“We found that dogs from both breed-types equally relied on body-awareness and social learning. They mainly opted for the detour (instead of using the shortcut) and less often approached the opening directly when the door was small plus they had observed the demonstrator before.”

 Introduction:  Overall the introduction is good and provides sufficient background references for the subject.  Additional clarification is needed in places, however.  See below for suggestions. 

Lines 59 and 61: These sentences should begin with the category of breed rather than breed examples.  This change would help with the flow of the paragraph.

RESPONSE: Thank you for the suggestion. We added a sentence to the very beginning of the paragraph with a short definition of “dog breed”. It reads like this (lines 96-99):
“Dog breeds are human-made, cultural and biological constructs. On one hand, breeds are defined by their official description (the ‘standard’); and on the other hand, a breed consists of specimens of a highly homogenous strain, characterized by breed-defining phenotypic traits and genomic makeup [Parker, 2012].”

Lines 116-119: Break this sentence into two sentence to improve clarity of description.

RESPONSE: Thank you for the request, we separated the long sentence into two and elaborated the text a little (lines 178-180). It reads now like this:

“Two solutions were offered: a longer detour and then a more optimal shortcut, which were eventually simultaneously available for the subjects. According to the protocol, at first, dogs had to repeatedly detour the obstacle while the shortcut (an opening through the barrier) was shut.”

Materials and Methods:  Overall, good.  Although it is mentioned early in the testing procedure, it would help to continue to clarify that the the methods described are identical to the study by Dobos and Pongracz (use citations).

RESPONSE: Thank you for the suggestion. We added now the reference (Dobos and Pongrácz, 2025) of the earlier study also to the subchapter ‘Testing equipment’ and to Table 2 (behavioural coding), to show the similarity of the methodologies.

Line 155: Please clarify from where the standards are sourced and how the classification is categorized (meaning what organization or authority provides the classification and distinction of breeds that was used to sort the dogs for this study).

RESPONSE: The official breed standards can be found at the FCI’s website. Now we indicated this in the text (lines 223-227):

“We did the group-assignment of the breeds according to the official description of the breeds in the standards (as they appear at the FCI (Fédération Cynologique Internationale) https://www.fci.be/en/Nomenclature/). The same method of breed assignment was previously used in several other publications (e.g., [Gácsi et al., 2009; Dobos and Pongrácz, 2024]).”

Lines 209-213: Please provide reasons for the experimenter to play with the dog or give the dog treats prior to the experiment. 

RESPONSE: Thank you for the comment. We added the explanation to this detail (lines 287-290), it reads like this:

“To ensure that the dog was well-motivated in the chosen reward, before the actual testing would start, the Experimenter gave a piece of treat to the dog from the plate we used for the experiment. In the case of toys, the Experimenter briefly played with the dog by using the toy provided by the Owner.”

Lines 232-238: In the control group description, the doors were closed for the first three trials.  Please clarify why the closure or opening of the doors was not varied in the control group.

RESPONSE: In the control groups we did not provide the dogs with a demonstration of detouring during the first three (closed door) trials. This was the only (and main) difference between the control and the test (detour demonstration) groups. Thus, how the doors were closed (in the first three trials) and then how they were opened (in Trials 4, 5 and 6) was identical in the control and testing (detour demonstration) conditions. In both conditions, there was a small door and a large door group. Accordingly, either the small, or the large door was used three times (in Trials 4, 5 and 6). To make the description clearer, we changed the wording a little at the Control groups’ section (lines 309-310):

The doors were always closed in the first 3 trials. Through the next three trials (Trial 4-6), we provided either a small, or a large opening to the dog, depending on the group-designation of the subject.

Section 2.8: Please clarify why the CI is important for this study.

RESPONSE: Thank you for this request. The rationale for using the CI has been already mentioned at the end of the Introduction. However, now we also added it to the indicated section of the Methods (lines 350-353):

“The head shape of the dogs can affect their visual performance with regard of the accuracy and ease of central and peripheral vision [9, 45-46]. Although we avoided testing strongly brachycephalic breeds, we wanted to add the head shape of the dogs as a potential confounder to the analysis.”

Results:  Overall the results are clear.  The statistical analyses were appropriate and the findings were presented in standard form. Visuals helped to clarify the findings and improve visualization of the results.

RESPONSE: We were pleased to read your supportive comment!

Discussion:  The discussion, limitations, and conclusions are appropriate and well written.  The discussion includes a lot more detail about brachycephalic breeds and justification for including CI.  I recommend including some of this in the introduction to help justify the measurements associated with CI and why it’s important. 

RESPONSE: Thank you for the suggestion and for your positive opinion about our article. We elaborated the end of the Introduction, now we state there that CI will be analyzed to account for its potential effect on dogs’ solution choice (lines 215-216):

“Lastly, in this study, we avoided testing strongly brachycephalic breeds (dogs with high Cephalic Index), because earlier it was found that due to their forward-facing eyes and specific retinal structure [45], these dogs can show biased responses to stimuli positioned right in front of them (e.g., human pointing signals: [9]); choice of openings to enter: [46]). Nevertheless, we took measurements of the subjects’ cephalic indices to be able to add this factor as a potential confounder to the analysis.”

The Limitations section should also discuss the difference between the experimenter being the social learning variable versus what might happen if the owner played this role. 

RESPONSE: Thank you for your request. We have added this paragraph to the Limitations (lines 661-670):

“Another factor that we did not test here was the familiarity of the demonstrator. In earlier experiments that measured dogs’ attentiveness and referential looking towards the nearby human, researchers found that dogs differentiated between their owner and a stranger [Mongillo et al., 2010; Merola et al., 2012]. However, in experiments where dogs had the opportunity to learn how to make a detour from either the owner or a stranger, no difference has been found between the effect of the demonstrators on dogs’ social learning [Pongrácz et al., 2001; 2021].”

Reviewer 2 Report

Comments and Suggestions for Authors

This is a well-written and well-organized paper that examines how body-size awareness, social learning (from a human demonstrator), and functional breed type (cooperative versus independent working dogs) influence problem solving in a spatial task. They found that body size-awareness and social learning influenced problem solving in both breed types; breed type influenced only time spent observing the demonstrator, with cooperative breeds spending more time than independent breeds.

My specific comments are detailed below, and many are editorial in nature.

Simple Summary and Abstract:

For readers unfamiliar with the difference between cooperative and independent working dog breeds it would be good to define each breed type. Currently, the distinction is not made until the Introduction (lines 57-63), which seems too late. For example, cooperative could be misconstrued for cooperating with other dogs (rather than humans) in working situations. This seems especially important in the Simple Summary, which is geared toward lay readers. However, please consider also doing this in the Abstract as well because some people may only read the Abstract.

Keywords: The keyword “dog” is in the title so will be picked up by searches; consider replacing “dog” with "detour test" or "detour task" or something similar that describes the test used.

Introduction:

It is good that the authors provided hypotheses tested and predictions made but I have a few questions.

Lines 130-136: The first prediction relates to a previous study on mixed breeds (ref 42) in comparison to working breeds (current study). Were the experimental conditions in the two studies identical (other than time of each study)? If so, it would be good to state this.

Lines 144-147: Here, the authors describe excluding dogs based on Cephalic Index (CI). But they do not mention that they measured CI of dogs included in the current study (see lines 206-208 and section 2.8), not for purposes of exclusion but for testing hypotheses related to reward latencies (lines 367-370 and Figure 7). This should be clarified in the last paragraph of the Introduction and a hypothesis provided for CI and reward latency.

Materials and methods:

Figure 1 is excellent. However, it might benefit from adding the original position of the dog (the symbol, O; lines 216-217) in the figure and legend.

Strengths of the study include that the reliability of coding was checked with an independent observer and the clear descriptions of behavioral variables in Table 2.

Sufficient detail is provided to allow replication.

Line 292 and elsewhere in the text: I suggest replacing “keeping” with “housing conditions” throughout the text because “housing conditions” will be clearer to most readers and was the first term used (line 162).

The statistical analyses are clearly presented and appear appropriate.

Results:

With only a few exceptions results are clearly presented: 1) Legend for Figure 2 – please check the last phrase for accuracy because it seems to contradict the above text (lines 310-311) and what is shown in Figure 2; 2) Legend for Figure 3 – For clarity (especially since some readers may only skim the figures and not read the full text), I would insert “(long or short)” after “cephalic index” and “(small or large)” after “door.”; 3) Figure 5 – Please increase the font of all axis labels.

Figure 8 shows Cephalic Indices of all participating dogs. Would this figure work better in the Materials and methods, where CI is described (section 2.8) or before the current Figure 7?

Discussion:

Section 4.1: Would it be possible to change the subsection title to “Strengths and Limitations of the Study” and add study strengths?

Conclusions:

I was confused about the comments regarding relative complexity of the task used in this study. For example, in line 555 it is described as “complex” (and earlier in lines 13-14, 130-132) while in other places it seemed to be described as simpler than some other designs (e.g., lines 499-506). Can this be clarified here and in other sections?

I enjoyed reading this paper and hope the authors find my comments helpful.

Author Response

Responses to REVIEWER 2

This is a well-written and well-organized paper that examines how body-size awareness, social learning (from a human demonstrator), and functional breed type (cooperative versus independent working dogs) influence problem solving in a spatial task. They found that body size-awareness and social learning influenced problem solving in both breed types; breed type influenced only time spent observing the demonstrator, with cooperative breeds spending more time than independent breeds.

RESPONSE: Dear Reviewer, we are especially thankful for your supportive and excellent comments. We incorporated the suggested changes and provide our point-by-point answers to your questions below.

My specific comments are detailed below, and many are editorial in nature.

Simple Summary and Abstract:

For readers unfamiliar with the difference between cooperative and independent working dog breeds it would be good to define each breed type. Currently, the distinction is not made until the Introduction (lines 57-63), which seems too late. For example, cooperative could be misconstrued for cooperating with other dogs (rather than humans) in working situations. This seems especially important in the Simple Summary, which is geared toward lay readers. However, please consider also doing this in the Abstract as well because some people may only read the Abstract.

RESPONSE: Thank you for your thoughtful suggestion. Although the word limits are strict, we managed to insert the requested information into both abstracts (lines 11-14 and 26-28). They read as:

(Simple summary)

“Hundreds of recognized dog breeds show exceptional variety of morphology, and often markedly different behaviors, too. Based on the extent of their reliance on human-given signals, many dog breeds are characterized as either cooperative or independent workers.”

(Abstract)

“Cooperative and independently working dog breeds differ in the extent of their reliance on human-given instructions, thus, they are ideal subjects for investigating dog-human interactions in a biologically relevant way.”

Keywords: The keyword “dog” is in the title so will be picked up by searches; consider replacing “dog” with "detour test" or "detour task" or something similar that describes the test used.

RESPONSE: Thank you for this very useful advice (we have never thought about this). We replaced “dog” with “detour task” in the keywords.

Introduction:

It is good that the authors provided hypotheses tested and predictions made but I have a few questions.

Lines 130-136: The first prediction relates to a previous study on mixed breeds (ref 42) in comparison to working breeds (current study). Were the experimental conditions in the two studies identical (other than time of each study)? If so, it would be good to state this.

RESPONSE: We agree. We added the requested information to the text where we describe the main goal of the current study (lines 172-173):

“By using the same protocol as the researchers with the mixed breed dogs did [42], we wanted to see whether purebred dogs from independent and cooperative types would behave similarly to mixed breeds, or they would respond differently, perhaps because they have a different selection past regarding tasks that include interactions with humans.”

Lines 144-147: Here, the authors describe excluding dogs based on Cephalic Index (CI). But they do not mention that they measured CI of dogs included in the current study (see lines 206-208 and section 2.8), not for purposes of exclusion but for testing hypotheses related to reward latencies (lines 367-370 and Figure 7). This should be clarified in the last paragraph of the Introduction and a hypothesis provided for CI and reward latency.

RESPONSE: Thank you for the relevant comment – this issue has been raised also by the other Reviewer. Accordingly, we added a sentence to the end of the Introduction where we now mention that CI measurements were collected and analyzed (lines 215-216), and we also elaborated further the necessity of this analysis in the methods (lines 350-353):

“Nevertheless, we still took measurements of the subjects’ cephalic indices for to be able to add this factor as a potential confounder to the analysis.”

“The head shape of the dogs can affect their visual performance with regard of the accuracy and ease of central and peripheral vision [9, 45-46]. Although we avoided testing strongly brachycephalic breeds, we wanted to add the head shape of the dogs as a potential confounder to the analysis.”

Materials and methods:

Figure 1 is excellent. However, it might benefit from adding the original position of the dog (the symbol, O; lines 216-217) in the figure and legend.

RESPONSE: Thank you for the suggestion – accordingly, we marked the “Start” position on Figure 1.

Strengths of the study include that the reliability of coding was checked with an independent observer and the clear descriptions of behavioral variables in Table 2.

Sufficient detail is provided to allow replication.

RESPONSE: Thank you very much, we are pleased to read this!

Line 292 and elsewhere in the text: I suggest replacing “keeping” with “housing conditions” throughout the text because “housing conditions” will be clearer to most readers and was the first term used (line 162).

RESPONSE: Thank you for the note, we fully agree. We replaced “keeping” with “housing” throughout the text.

The statistical analyses are clearly presented and appear appropriate.

RESPONSE: Thank you, we are very happy to read this.

Results:

With only a few exceptions results are clearly presented: 1) Legend for Figure 2 – please check the last phrase for accuracy because it seems to contradict the above text (lines 310-311) and what is shown in Figure 2;

RESPONSE: Thank you for noticing our mistake in the figure legend, we corrected it now (lines 414-416). Dogs opted for the detour in Trial 4, BEFORE and not AFTER repeated exposure to open doors:

Figure 3. Solution choice during the open-door trials (trials 4-6), in the experimental groups. Dogs opted more often to make detours when they faced the small door; and in Trial 4, before repeated exposure to the open doors.”

 2) Legend for Figure 3 – For clarity (especially since some readers may only skim the figures and not read the full text), I would insert “(long or short)” after “cephalic index” and “(small or large)” after “door.”;

RESPONSE: Thank you for the careful suggestion, we added the requested information to the legend of this figure.

3) Figure 5 – Please increase the font of all axis labels.

RESPONSE: We have increased the font size of the axis labels as you requested.

Figure 8 shows Cephalic Indices of all participating dogs. Would this figure work better in the Materials and methods, where CI is described (section 2.8) or before the current Figure 7?

RESPONSE: Thank you for this suggestion, we agree with you that this Figure was somewhat ‘out of context’ here. We moved the Figure and the corresponding text belonging to the CI analysis to the beginning of the Results. Thus, Figure 8 became Figure 2, and all the subsequent figures have been re-numbered accordingly.

Discussion:

Section 4.1: Would it be possible to change the subsection title to “Strengths and Limitations of the Study” and add study strengths?

RESPONSE: Thank you for this very interesting and kind suggestion. We think that this would be quite unusual compared to most publications where authors list only limitations of their study. We would like to believe that the strengths of our study become obvious by reading the results and conclusions. We would like to keep the ‘Limitations’ section therefore intact (actually, by following the request of the other Reviewer, we added more details to the limitations…).

Conclusions:

I was confused about the comments regarding relative complexity of the task used in this study. For example, in line 555 it is described as “complex” (and earlier in lines 13-14, 130-132) while in other places it seemed to be described as simpler than some other designs (e.g., lines 499-506). Can this be clarified here and in other sections?

RESPONSE: Thank you for this question. When we write about “complex task”, we always mean the combination of three factors: detour; social learning (via observing the demonstration); and body-awareness-based option for a shortcut. The ‘simple’ or ‘easier’ adjectives refer to the apparently easier nature of the straight obstacle (to detour) compared to the more often used V-shaped one. To make this issue clearer, we added text to the Conclusions (lines 684-687 and 691-694):
“One explanation might be that in such complex tasks as the one we used here (detour with or without demonstration, plus a shortcut with more or less convenient door size), flexible behaviour provides the strongest advantage, and such flexibility can be achieved by learning (i.e., body-awareness and social learning) rather than by directional selection”

“The relatively easy task (making a detour around a straight obstacle, using a shortcut) could also contribute to this outcome, as it was found earlier that cooperative and independent breeds did not differ in their persistence and capabilities, in individual problem-solving tasks”

I enjoyed reading this paper and hope the authors find my comments helpful.

RESPONSE: Thank you very much for the useful and polite advice, suggestions. We appreciate your efforts a lot!